# Predictive role of preoperative sarcopenia for long-term survival in rectal cancer patients: A meta-analysis

Qiutong Su [1]*, Jia Shen [2]

1 Department of Gastroenterology and Hepatology, West China Hospital, Sichuan University/West China School of Nursing, Sichuan University, Chengdu, China, 2 Department of Critical Care Medicine, West China Hospital, Sichuan University/West China School of Nursing, Sichuan University, Chengdu, China

* ss5323001@163.com

**Data Availability Statement:** All data generated or analyzed during this study are included in this published article.

**Funding:** The author(s) received no specific funding for this work.

## Abstract

### Purpose

To identify the predictive role of sarcopenia in long-term survival among rectal cancer patients who underwent surgery based on available evidence.

### Methods

The Medline, EMBASE and Web of Science databases were searched up to October 20, 2023, for relevant studies. Overall survival (OS), disease-free survival (DFS) and cancer-specific survival (CSS) were the endpoints. Hazard ratios (HRs) and 95% confidence intervals (CIs) were combined to evaluate the association between sarcopenia and survival.

### Results

Fifteen studies with 4283 patients were included. The pooled results demonstrated that preoperative sarcopenia significantly predicted poorer OS (HR = 2.07, 95% CI = 1.67–2.57, P<0.001), DFS (HR = 1.85, 95% CI = 1.39–2.48, P<0.001) and CSS (HR = 1.83, 95% CI = 1.31–2.56, P<0.001). Furthermore, subgroup analysis based on neoadjuvant therapy indicated that sarcopenia was a risk factor for worse OS and DFS in patients who received (OS: HR = 2.44, P<0.001; DFS: HR = 2.16, P<0.001) but not in those who did not receive (OS: HR = 2.44, P<0.001; DDFS: HR = 1.86, P = 0.002) neoadjuvant chemoradiotherapy. In addition, subgroup analysis based on sample size and ethnicity showed similar results.

### Conclusion

Preoperative sarcopenia is significantly related to poor survival in surgical rectal cancer patients and could serve as a novel and valuable predictor of long-term prognosis in these patients.

**Competing interests:** The authors have declared that no competing interests exist.

## Introduction

Colorectal cancer is one of the most common malignant tumors of the digestive system in China [1, 2]. According to reports, in 2015, there were 387,600 new cases of colorectal cancer in China, ranking fourth in incidence among all malignant tumors. Colorectal cancer accounts for 8.01% of all cancer deaths in China, ranking fifth in the spectrum of cancer deaths and making it one of the leading causes of malignant tumor-related deaths in the country [3]. Compared to those in foreign countries, colorectal cancer in China is more commonly located in the rectum [4]. Generally, surgery is the primary treatment for rectal cancer patients. However, accurately predicting postoperative survival has always been a challenging issue in clinical practice.

Malnutrition is a common issue among cancer patients, and many tumor patients inevitably experience weight loss, which has adverse effects on cancer treatment and prognosis [5, 6]. Weight loss may result from reduced nutritional intake, such as anorexia, or from tumor-related consumption. Additionally, the side effects of antitumor treatments can exacerbate this condition [7, 8]. Sarcopenia refers to a progressive and generalized skeletal muscle disorder associated with increased risks of adverse outcomes such as falls, fractures, physical disability, and mortality. Research has confirmed a close association between sarcopenia and various chronic diseases, including chronic obstructive pulmonary disease (COPD), chronic heart failure, and chronic kidney disease [9, 10]. The diagnosis of sarcopenia primarily involves assessing muscle mass, muscle strength, and physical performance. The final confirmation is made by determining the presence of low muscle mass. Furthermore, studies indicate that preexisting sarcopenia is prevalent among cancer patients and can adversely affect their prognosis. It has been reported that sarcopenia can lead to poor prognosis in several types of cancers, such as lung cancer, esophageal cancer and gastric cancer [11–13]. However, for now, the association between sarcopenia and long-term prognosis in rectal cancer patients who underwent surgery has not been determined.

Therefore, the aim of this meta-analysis was to further clarify the predictive role of preoperative sarcopenia for long-term survival among surgical rectal cancer patients.

## Materials and methods

This meta-analysis was conducted according to the Preferred Reporting Items for Systematic Review and Meta-Analyses 2020 [14].

### Ethical approval

All procedures performed in studies involving human participants were in accordance with the ethical standards of the institutional and/or national research committee and with the 1964 Helsinki Declaration and its later amendments or comparable ethical standards. For this type of study, formal consent is not needed.

### Literature search

The Medline, EMBASE and Web of Science databases were searched from inception to October 20, 2023. In this meta-analysis, the following terms were used during the search: sarcopenia, rectal, rectum, cancer, tumor, neoplasm, carcinoma, surgery, resection, preoperative, prognosis, prognostic and survival. MeSH terms and free texts were applied. The specific search strategies used for each database are presented in S1 File.

### Inclusion criteria

Studies that met the following criteria were included in our meta-analysis: 1) patients were diagnosed with primary rectal cancer and received surgical therapy; 2) sarcopenia was defined according to the skeletal muscle index (SMI), psoas muscle mass index (PMI) or other similar indicators before surgery; 3) patients were divided into sarcopenia and nonsarcopenia groups, and overall survival (OS), disease-free survival (DFS) and (or) cancer-specific survival (CSS) were compared between groups; 4) hazard ratios (HRs) with 95% confidence intervals (CIs) for OS, DFS and CSS were reported; and 5) studies were published in English.

### Exclusion criteria

The exclusion criteria for studies were as follows: 1) insufficient, duplicated or overlapped data; 2) editorials, animal trials, case reports or reviews; and 3) studies that did not report HRs with 95% CIs for survival, including studies providing Kaplan–Meier survival curves because of significant bias in the calculation of HRs with 95% CIs from Kaplan–Meier survival curves.

### Data collection

We collected the following information from each included study: the first author, publication year, sample size, country, TNM stage, history of neoadjuvant therapy and adjuvant therapy, definition of sarcopenia, endpoint, follow-up period, HR with corresponding 95% CI and source of HRs (multivariate analysis or univariate analysis).

### Quality assessment

All included studies were cohort studies; thus, the methodological quality was evaluated according to the Newcastle–Ottawa Scale (NOS) score, and studies with NOS scores >5 were defined as high-quality studies [15].

### Statistical analysis

All analyses were performed with STATA version 15.0 software. The heterogeneity between studies was calculated by $I^2$ statistics and the Q test. If significant heterogeneity was detected, represented by $I^2 > 50\%$ and/or $P < 0.1$, the random effects model was applied; otherwise, the fixed effects model was applied [16]. The HRs with 95% CIs from multiple regression models were applied whenever available. The hazard ratio (HR) and 95% confidence interval (CI) were combined to evaluate the relationship between preoperative survival and survival. Subgroup analysis based on the history of neoadjuvant therapy, sample size and ethnicity was performed. Sensitivity analysis was performed to detect the sources of heterogeneity and assess the stability of the pooled results. In addition, Begg's funnel plot and Egger's test were conducted to detect publication bias, and significant publication bias was defined as $P < 0.05$ [17, 18]. If obvious publication bias was detected, then the fill-and-trim method was applied to identify potentially unpublished studies [19].

## Results

### Literature search

As shown in Fig 1, 214 records were found in these three databases (PubMed: n = 54, EMBASE: n = 77 and Web of Science: n = 83), and 39 duplicated records were removed. After

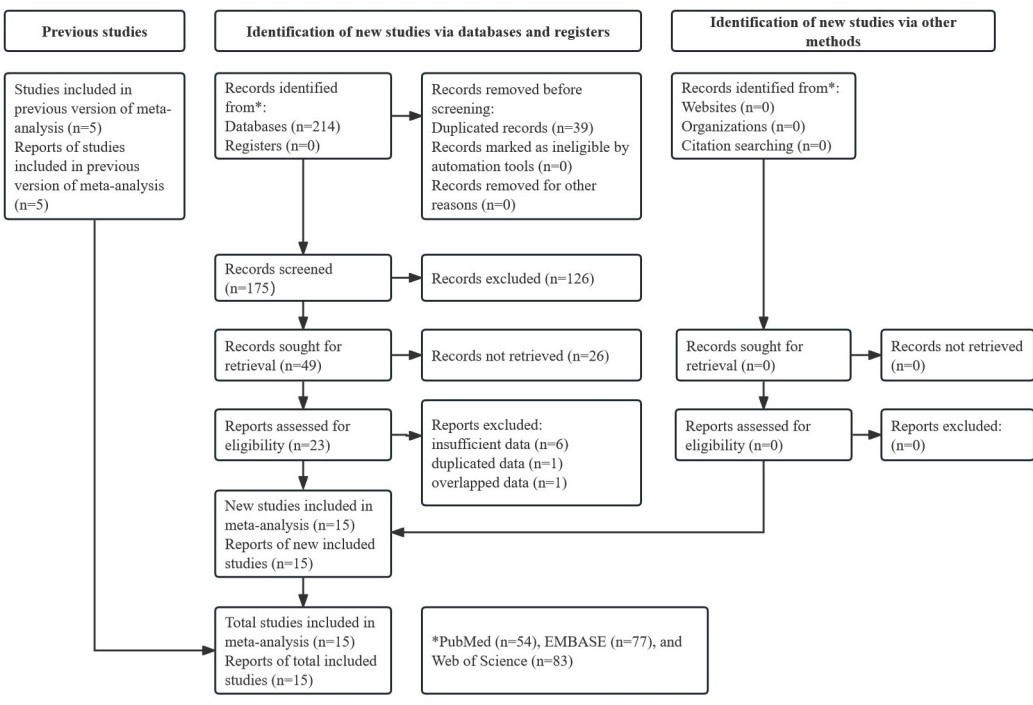

**Fig 1. Prisma flow diagram of this meta-analysis.**

reviewing the titles, 126 records were excluded. Then, 26 additional publications were excluded after reviewing the abstracts. After carefully reviewing the full texts, 15 studies were eventually included in this meta-analysis [20–34].

## Basic characteristics of the included studies

Detailed information about the included studies is presented in Table 1. All included studies were retrospective. A total of 4283 patients were enrolled, and the sample size ranged from 46 to 1384 patients. Most related studies were from Asian countries, including China, Korea, South Korea and Japan, and sarcopenia was evaluated according to the SMI in most studies. In addition, all included studies were high-quality studies with an NOS score ≥6.

## Predictive role of preoperative sarcopenia for OS in rectal cancer patients

Thirteen studies explored the relationship between sarcopenia and OS among surgical rectal cancer patients [20–29, 31, 32, 34]. The pooled results demonstrated that the presence of preoperative sarcopenia was significantly related to poor OS (hazard ratio (HR) = 2.07, 95% CI = 1.67–2.57, P<0.001; $I^2$ = 39.5%, P = 0.071) (Fig 2). Subgroup analysis stratified by history of neoadjuvant therapy revealed that both patients with (HR = 2.44, 95% CI = 1.54–3.87, P<0.001; $I^2$ = 0.0%, P = 0.432) [20, 23, 28, 32] and without (HR = 2.44, 95% CI = 1.74–3.42, P<0.001; $I^2$ = 0.0%, P = 0.561) [22, 27, 29] neoadjuvant chemoradiotherapy-related sarcopenia had worse OS. In addition, subgroup analyses based on sample size (<200 vs. ≥200) and ethnicity (Asian vs. non-Asian) showed similar results (Table 2).

**Table 1. Basic characteristics of included studies.**

| Author | Year | Sample size | Country | TNM stage | Neoadjuvant therapy | Adjuvant therapy | Definition of sarcopenia | Endpoint | Follow-up time | Source of HRs | NOS |
|---|---|---|---|---|---|---|---|---|---|---|---|
| Choi [20] | 2018 | 188 | South Korea | II-III | Chemoradiotherapy | NR | Men: SMI<52.4cm²/m², Women: SMI<38.5cm²/m² | OS, DFS | 5–91 months | M | 7 |
| Park [21] | 2018 | 104 | South Korea | I-III | Mixed | Mixed | Men: SMI<49cm²/m², Women: SMI<31cm²/m² | OS, DFS | 106.8 (median) months | OS: M, DFS: U | 7 |
| Hopkins [22] | 2019 | 381 | Canada | I-III | No | Mixed | Men: SMI<43cm²/m², Women: SMI<41cm²/m² | OS, DFS, CSS | 5.2 (median) (0.01–10.25) years | M | 7 |
| Chung [23] | 2020 | 93 | South Korea | I-III | Chemoradiotherapy | Mixed | Men: SMI<52.4cm²/m², Women: SMI<38.5cm²/m² | OS | NR | U | 6 |
| Han [24] | 2020 | 1384 | Korea | I-III | Mixed | Mixed | Men: SMI<52.4cm²/m², Women: SMI<38.5cm²/m² | OS, DFS | NR | U | 8 |
| Shirdel [25] | 2020 | 264 | Sweden | I-III | Mixed | Mixed | Men: SMI<49.2cm²/m², Women: SMI<38.1cm²/m² | OS, CSS | 6.2 (median) (4.7–10.9) years | M | 7 |
| Wang [26] | 2020 | 212 | China | I-III | NR | NR | Men: SMI<38.89cm²/m², Women: SMI<33.28cm²/m² | OS, DFS | 63 (median) (6–80) months | M | 7 |
| Xie [27] | 2020 | 152 | China | NR | No | Mixed | Men: SMI<49.5cm²/m², Women: SMI<29.9cm²/m² | OS, DFS | 62 (median) (1–80) months | M | 7 |
| Abe [28] | 2022 | 225 | Japan | I-III | Chemoradiotherapy | Mixed | Men: PMI<5.32cm²/m², Women: PMI<4.11cm²/m² | OS | 73.4 months (median) | M | 6 |
| Giani [29] | 2022 | 129 | Italy | I-IV | No | Mixed | Men: SMI<45.9cm²/m², Women: SMI<38.7cm²/m² | OS, DFS, CSS | 96.7 (median) (61.6–119.5) months | U | 6 |
| Horie [30] | 2022 | 46 | Japan | I-III | Chemoradiotherapy | Mixed | Men: PMV<140.93cm³/m², Women: PMV<105.8cm³/m² | DFS | NR | M | 7 |
| Abe [31] | 2023 | 708 | Japan | I-IV | Mixed | Mixed | Men: PMI<6.36cm²/m², Women: PMI<3.92cm²/m² | OS, DFS, CSS | 61.1 (median) (36.3–86.9) months | M | 8 |
| Gartrell [32] | 2023 | 132 | Austria | II-III | Chemoradiotherapy | Mixed | Men: SMI<47.5cm²/m², Women: SMI<39.1cm²/m² | OS, DFS | 62.8 months (mean) | M | 7 |
| Mallet-Boutboul [33] | 2023 | 100 | France | NR | Chemoradiotherapy | Mixed | Men: SMI<52.4cm²/m², Women: SMI<38.5cm²/m² | DFS | 33.9 (median) (15.460.4) months | M | 7 |
| Portale [34] | 2023 | 165 | Italy | I-III | Mixed | NR | Men: SMI<52.4cm²/m², Women: SMI<38.5cm²/m² | OS, DFS | 69 (median) (40.7–100.4) months | OS: M, DFS: U | 7 |

NR: not reported; TNM: tumor-node-metastasis; SMI: skeletal muscle index; PMI: Psoas muscle mass index; PMV: l psoas muscle volume; OS: overall survival; DFS: disease-free survival; CSS: cancer-specific survival; M: multivariate analysis; U: univariate analysis; NOS: Newcastle-Ottawa Scale.

### Predictive role of preoperative sarcopenia for DFS in rectal cancer patients

Twelve studies reported the association between sarcopenia and DFS in surgical rectal cancer patients [20–22, 24, 26, 27, 29–34]. The pooled results indicated that preoperative sarcopenia significantly predicted poor DFS (hazard ratio (HR) = 1.85, 95% CI = 1.39–2.48, P<0.001; $I^2$ = 58.5%, P = 0.005) (Fig 3). Moreover, subgroup analysis based on the history of neoadjuvant therapy revealed similar results (patients who received neoadjuvant chemoradiotherapy: HR = 2.16, 95% CI = 1.42–3.31, P<0.001; $I^2$ = 0.0%, P = 0.485 [20, 30, 32, 33]; patients who did not receive neoadjuvant chemoradiotherapy: HR = 1.86, 95% CI = 1.25–2.75, P = 0.002; $I^2$ = 30.8%, P = 0.236 [22, 27, 29]). In addition, subgroup analyses based on sample size (<150 vs. ≥150) and ethnicity (Asian vs. non-Asian) produced similar results (Table 2).

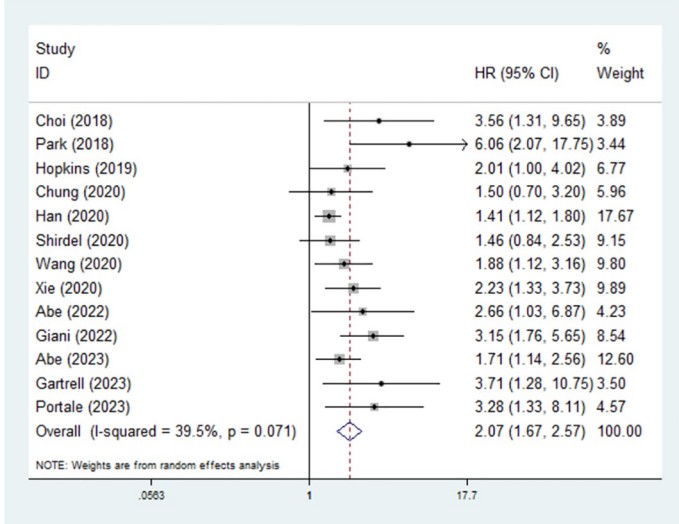

**Fig 2. Predictive role of preoperative sarcopenia for overall survival in rectal cancer patients.**

**Table 2. Results of pooled analysis.**

| | No. of studies | HR | 95% CI | P value | I² (%) | P value |
|---|---|---|---|---|---|---|
| Overall survival | 13 [20–29, 31, 32, 34] | 2.07 | 1.67–2.57 | <0.001 | 39.5 | 0.071 |
| Neoadjuvant therapy | | | | | | |
| Chemoradiotherapy | 4 [20, 23, 28, 32] | 2.44 | 1.54–3.87 | <0.001 | 0.0 | 0.432 |
| No | 3 [22, 27, 29] | 2.44 | 1.74–3.42 | <0.001 | 0.0 | 0.561 |
| Sample size | | | | | | |
| <200 | 7 [20, 21, 23, 27, 29, 32, 34] | 2.74 | 2.07–3.65 | <0.001 | 1.3 | 0.414 |
| ≥200 | 6 [22, 24–26, 28, 31] | 1.58 | 1.33–1.88 | <0.001 | 0.0 | 0.685 |
| Ethnicity | | | | | | |
| Asian | 8 [20, 21, 23, 24, 26–28, 31] | 1.72 | 1.45–2.03 | <0.001 | 41.2 | 0.104 |
| Non-Asian | 5 [22, 25, 29, 32, 34] | 2.30 | 1.69–3.14 | <0.001 | 23.7 | 0.264 |
| Disease-free survival | 12 [20–22, 24, 26, 27, 29–34] | 1.85 | 1.39–2.48 | <0.001 | 58.5 | 0.005 |
| Neoadjuvant therapy | | | | | | |
| Chemoradiotherapy | 4 [20, 30, 32, 33] | 2.16 | 1.42–3.31 | <0.001 | 0.0 | 0.485 |
| No | 3 [22, 27, 29] | 1.86 | 1.25–2.75 | 0.002 | 30.8 | 0.236 |
| Sample size | | | | | | |
| <150 | 5 [21, 29, 30, 32, 33] | 2.84 | 1.87–4.32 | <0.001 | 0.0 | 0.787 |
| ≥150 | 7 [20, 22, 24, 26, 27, 31, 34] | 1.54 | 1.13–2.10 | 0.006 | 57.7 | 0.028 |
| Ethnicity | | | | | | |
| Asian | 7 [20, 21, 24, 26, 27, 30, 31] | 1.87 | 1.28–2.75 | 0.001 | 72.2 | 0.001 |
| Non-Asian | 5 [22, 29, 32–34] | 1.86 | 1.24–2.80 | 0.003 | 0.0 | 0.457 |
| Cancer-specific survival | 4 [22, 25, 29, 31] | 1.83 | 1.31–2.56 | <0.001 | 0.0 | 0.514 |

HR: hazard ratio; CI: confidence interval.

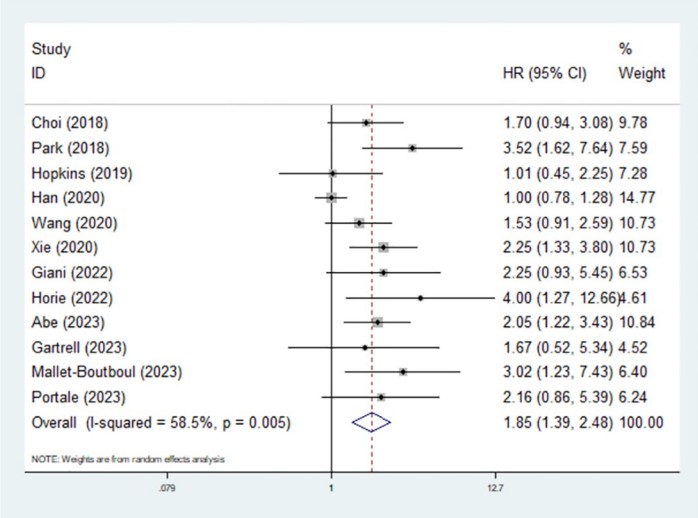

**Fig 3. Predictive role of preoperative sarcopenia for disease-free survival in rectal cancer patients.**

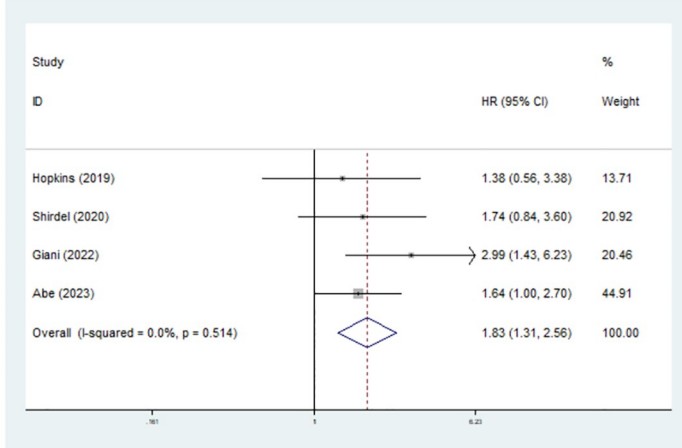

**Fig 4. Predictive role of preoperative sarcopenia for cancer-specific survival in rectal cancer patients.**

## Predictive role of preoperative sarcopenia for CSS in rectal cancer patients

Only four studies have explored the association between sarcopenia and postoperative CSS in rectal cancer patients [22, 25, 29, 31]. The pooled results verified that preoperative sarcopenia also predicted worse CSS (hazard ratio (HR) = 1.83, 95% CI = 1.31–2.56, P<0.001; $I^2$ = 0.0%, P = 0.514) (Fig 4).

## Sensitivity analysis

Sensitivity analysis of the relationship between preoperative sarcopenia incidence and OS and DFS was performed. According to Fig 5A and 5B, the pooled results of this meta-analysis were stable and reliable, and none of the included studies caused an obvious impact on the overall results.

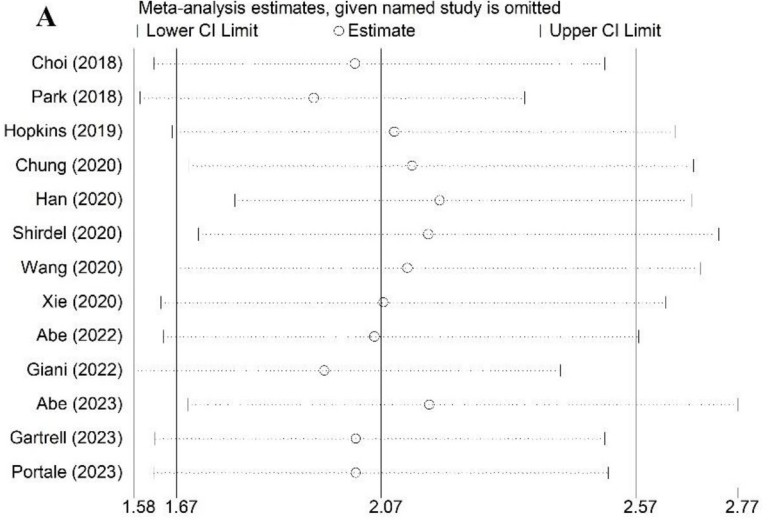

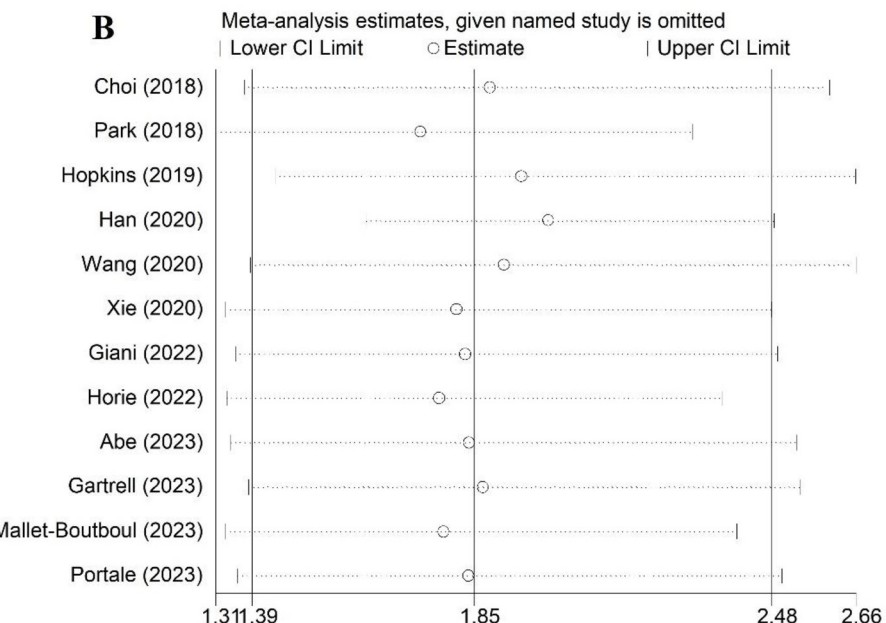

**Fig 5. Sensitivity analysis for the association of preoperative sarcopenia with overall survival (A) and disease-free survival (B) in rectal cancer patients.**

## Publication bias

According to Begg's funnel plots (Fig 6A and 6B) and Egger's test (P = 0.001 and P = 0.003) for the associations between preoperative sarcopenia and OS and DFS, significant publication bias was detected. Thus, the fill-and-trim method was applied, and five (Fig 6C) and four (Fig 6D) potentially unpublished studies were found. However, five (OS: filled HR = 1.66, 95% CI = 1.31–2.10, P<0.001) and four (DFS: filled HR = 1.56, 95% CI = 1.19–2.03, P = 0.001) "unpublished studies" did not affect the overall conclusions.

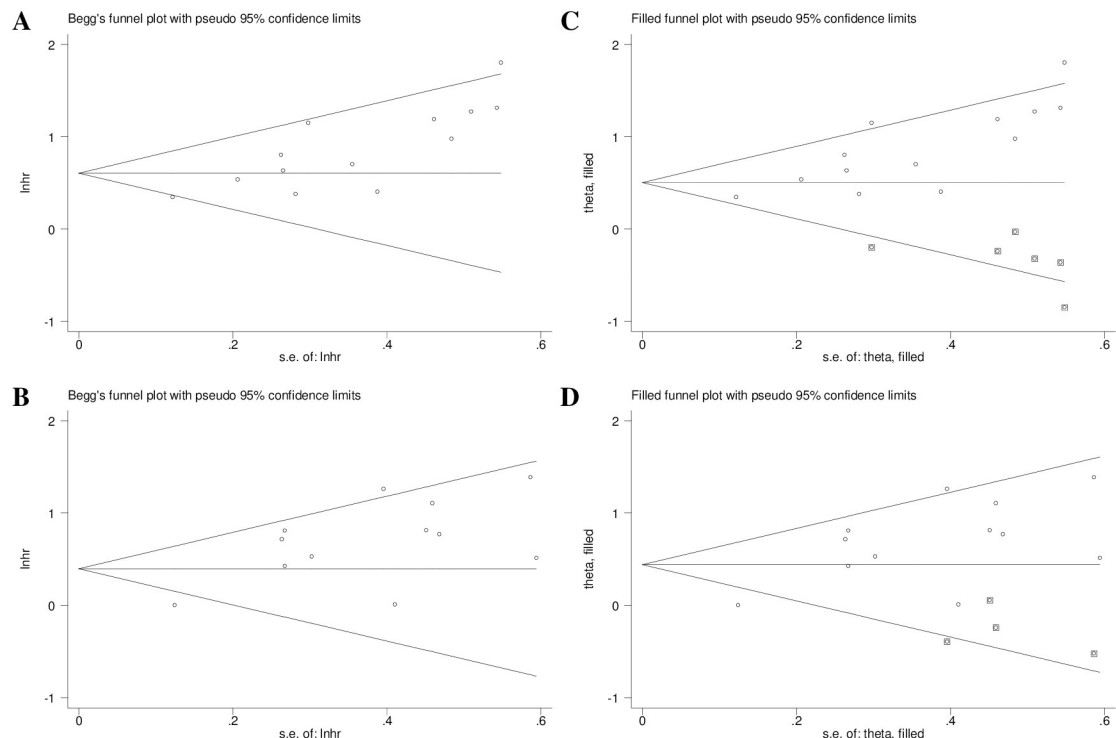

**Fig 6. Begg's and filled funnel plots for the association of preoperative sarcopenia with overall survival (A and B) and disease-free survival (C and D) in rectal cancer patients.**

## Discussion

In this meta-analysis, we demonstrated that sarcopenia was significantly associated with long-term survival among rectal cancer patients who received surgical therapy and that patients with sarcopenia had poorer survival. Furthermore, a history of neoadjuvant therapy did not significantly affect the association between sarcopenia and poor prognosis. Thus, sarcopenia could serve as a valuable prognostic factor in surgical-related rectal cancer patients. However, additional high-quality studies are still needed to further verify the above findings. Zhu et al. conducted a meta-analysis to determine the prognostic role of sarcopenia in rectal cancer patients [35]. These authors included seven studies with 2377 patients and reported that sarcopenia predicted poor OS (HR = 2.37, 95% CI = 1.13–4.98, P = 0.02) [35]. However, their findings are relatively limited. In this meta-analysis, 15 studies were included, and patients who underwent surgery were the focus. In addition, the associations of sarcopenia with postoperative DFS and CSS were also assessed, and subgroup analysis based on neoadjuvant chemoradiotherapy was further conducted. Therefore, this meta-analysis provides additional high-quality evidence about the prognostic role of sarcopenia in surgically treated rectal cancer patients.

The impact of muscle loss on the prognosis of cancer patients involves various factors. First, a reduction in muscle mass may lead to a decreased ability for postoperative recovery, resulting in reduced surgical tolerance. Surgery is a primary modality for treating many tumors, and a patient's postoperative recovery directly correlates with prognosis. Second, muscle loss can influence the functionality of the immune system. The immune system plays a crucial role in combating tumor cells and preventing infections. When immune function

decreases, patients are more susceptible to complications such as infections, thus affecting prognosis [36, 37]. Additionally, muscle loss may result in decreased tolerance to chemotherapy and radiation therapy. These treatments impose a significant burden on the body, and patients with weakened physical constitutions may struggle to endure side effects, impacting treatment completion and efficacy [11, 38]. Moreover, there is an association between muscle loss and systemic inflammation. Inflammation may play a role in the development of tumors and is related to systemic complications associated with malignancies, further influencing patient prognosis [39, 40]. Finally, muscle loss is often associated with malnutrition. Maintaining good nutritional status is crucial for a patient's recovery and treatment outcomes. Muscle loss may signify insufficient nutritional support, exacerbating malnutrition.

Our meta-analysis confirmed that sarcopenia is a risk factor affecting the prognosis of rectal cancer patients. Therefore, appropriate and reasonable interventions for sarcopenia may help improve the prognosis of rectal cancer patients. Currently, there are three main treatment options for sarcopenia: nutritional supplementation, exercise intervention, and drug therapy. Current research evidence indicates that nutritional therapy is beneficial for improving muscle mass, strength, and muscle function [41]. Key nutritional components include protein supplementation, creatine, vitamins, and omega-3 polyunsaturated fatty acids, among others, with protein supplementation being crucial for improving nutritional status [41]. In addition, proper physical exercise can increase muscle mass, strength, and endurance, thereby improving muscle wasting [42, 43]. It is currently believed that combining exercise with protein and energy intake is key to preventing and treating sarcopenia [42, 43]. Treatment for sarcopenia primarily includes testosterone, selective androgen receptor modulators, growth hormone analogs, etc. [44, 45]. However, there is a lack of standardized international guidelines for the systematic treatment of sarcopenia. Many intervention measures are still in the early exploration stage, so further research is needed to provide additional high-quality evidence-based support.

There are several limitations in our meta-analysis that should be noted. First, the overall sample size was relatively small, and all the studies were retrospective. Second, several confounding factors, such as the definition of sarcopenia, including the cutoff values of SMI, PMV and PMI and adjuvant therapy, were included in this meta-analysis, and we were unable to conduct a more detailed analysis due to the lack of original data. Third, most of the included studies were from Asian countries, which might affect the generalizability of our findings. Fourth, due to the lack of relevant data, we failed to explore the association between sarcopenia and postoperative short-term outcomes. Fifth, significant publication bias was detected in our meta-analysis. Although the results of the fill-and-trim method indicated that "unpublished studies" did not affect the overall findings, additional high-quality studies are still needed to verify our conclusions.

## Conclusion

Preoperative sarcopenia is significantly associated with poor survival among surgical rectal cancer patients and could serve as a novel and valuable predictor of long-term prognosis in these patients.

## Supporting information

**S1 File. Literature search strategies in each database.**
(DOCX)

**S1 Checklist. PRISMA 2020 checklist.**
(DOCX)

## Author Contributions

**Conceptualization:** Qiutong Su.

**Data curation:** Qiutong Su, Jia Shen.

**Formal analysis:** Jia Shen.

**Software:** Qiutong Su.

**Writing – original draft:** Qiutong Su.

**Writing – review & editing:** Jia Shen.

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
