## [Decision Letter · Decision Letter 0]

16 Jan 2024

PONE-D-23-38922Predictive role of preoperative sarcopenia for long-term survival in rectal cancer patients: a meta-analysisPLOS ONE

Dear Dr. Su,

Thank you for submitting your manuscript to PLOS ONE. After careful consideration, we feel that it has merit but does not fully meet PLOS ONE’s publication criteria as it currently stands. Therefore, we invite you to submit a revised version of the manuscript that addresses the points raised during the review process.

We look forward to receiving your revised manuscript.

Kind regards,

Zubing Mei, MD,PH.D

Academic Editor

PLOS ONE

Journal Requirements:

2. We note that your Data Availability Statement is currently as follows: All data generated or analyzed during this study are included in this published article.

Additional Editor Comments:

The manuscript presents valuable insights into the prognostic value of sarcopenia in rectal cancer. The reviewers have provided constructive feedback that warrants a major revision. Specifically pay attention the following points:

1. The manuscript requires thorough English editing to correct grammatical and spelling errors. This is crucial to ensure clear communication of the research findings.

2. Clarifications regarding the inclusion and exclusion criteria of studies, particularly the rationale behind excluding studies that provide Kaplan-Meier survival curves instead of HR with 95%, are necessary.

3. The association between sarcopenia and postoperative short-term outcomes should be explored or discussed as a limitation.

4. Detailed methodology and clarity in reporting, such as literature search strategy, study selection process, and handling of hazard ratios, are necessary for reproducibility and transparency.

5. Include citations for statements concerning survival analysis and the types of included studies (prospective, retrospective, or RCT), as well as follow-up time specificity in table 1.

6. Subgroup analyses based on predefined criteria and a discussion on publication bias are recommended to strengthen the paper's validity.

7. Discrepancies in the numbers presented in tables regarding OS and DFS need to be addressed and corrected for consistency and clarity.

8. All abbreviations, such as SMI, must be fully described upon their initial mention in the text, ensuring accessibility for all readers.

Reviewers' comments:

Reviewer's Responses to Questions

**Comments to the Author**

1. Is the manuscript technically sound, and do the data support the conclusions?

Reviewer #1: Yes

Reviewer #2: Yes

Reviewer #3: Yes

2. Has the statistical analysis been performed appropriately and rigorously? 

Reviewer #1: Yes

Reviewer #2: Yes

Reviewer #3: Yes

3. Have the authors made all data underlying the findings in their manuscript fully available?

Reviewer #1: Yes

Reviewer #2: Yes

Reviewer #3: No

4. Is the manuscript presented in an intelligible fashion and written in standard English?

Reviewer #1: Yes

Reviewer #2: Yes

Reviewer #3: Yes

5. Review Comments to the Author

Reviewer #1: Thanks for giving me the opportunity to review this work. In overall, this manuscript was well designed and written. It is a update meta-analysis investigating the prognostic value of sarcopenia in rectal cancer. I have few concerns about this work which would be addressed.

1. English editing is recommended and some grammar errors exist in the manuscript.

2. Studies providing the Kaplan-Meier survival curves instend of HR with 95% were excluded? Why?

3. The authors did not identify the association between sarcopenia and postoperative short-term outcomes in rectal cancer patients. It's also a limitation.

Reviewer #2: The authors included 15 studies and indicated the prognostic value of sarcopenia in rectal cancer. Congratulations on this successful manuscript. I have some minor suggestions for authors to further improve the quality of their work.

1. I would recommend suggest the author carefully check and modify the manuscript and some spelling mistakes and grammatical mistakes could be noticed in this paper.

2. In the "literature search" part, detailed process should be described.

3. "Thirteen studies explored the relationship between sarcopenia and OS among surgical rectal cancer patients." The references should be cited. The same suggestion for "Twelve studies manifested the association of sarcopenia with DFS among surgical rectal cancer patients." and "Only four studies explored the association of sarcopenia with postoperative CSS in rectal cancer patients."

4. What are the types of included studies? Retrospective or prospective? Or RCT?

5. In table 1, "mean" or "median" follow-up time should be specificly described for each studies.

Reviewer #3: In this study, the authors conducted a meta-analysis to evaluate the prognostic significance of preoperative sarcopenia in rectal cancer patients. Interestingly, they found that preoperative sarcopenia is associated with poor clinical outcomes in this disease. The manuscript's field is interesting, the introduction and discussion are well-structured; the result section is appropriate; however, there are some concerns, and major revision is expected.

1. To enhance the transparency of the methodology, the authors should consider presenting the literature search strategy employed in various databases in a table. Furthermore, it is important for them to explicitly state the number of studies obtained from each database.

2. For a more comprehensive description of the inclusion and exclusion criteria, it is advisable to provide a thorough explanation that encompasses all relevant aspects. This should include any limitations on language or excluding review articles.

3. In the methods section, the authors should mention whether all hazard ratios (HRs) were provided in the included papers based on univariate or multivariate analysis (in table 1, add which one was for each study) or if they calculated the HRs based on the Kaplan-Meier curves.

4. In the method section, on page 5, lines 1-2, please replace the word 'or' with 'otherwise' (or any suitable synonym) in the following sentence: 'The random effects model was applied; or the fixed effects model was applied.' The use of 'or' is incorrect.

5. I recommend doing subgroup analysis based on different criteria such as sample size (based on mean or median), Newcastle–Ottawa Scale score (<7 and >7), ethnicity (Asian and non-Asian), and so on.

6. In the discussion part, specially limitation section, the authors should mention and interpret the results regarding publication bias.

7. In the table 2, the overall number of studies for OS is 13. However, the number of studies for Chemoradiotherapy and NO subgroups are written 4 and 3, respectively. It is unclear. Because 4 plus 3 will be 7, not 13. So, please correct these numbers for OS and, also, DFS.

8. Abbreviations such as SMI (Page 5) should be fully described when initially mentioned.

6. PLOS authors have the option to publish the peer review history of their article (what does this mean?). If published, this will include your full peer review and any attached files.

Reviewer #1: No

Reviewer #2: No

Reviewer #3: No

---

## [Author Response · Author response to Decision Letter 0]

22 Jan 2024

Response to editor:

The manuscript presents valuable insights into the prognostic value of sarcopenia in rectal cancer. The reviewers have provided constructive feedback that warrants a major revision. Specifically pay attention the following points:

Question 1. The manuscript requires thorough English editing to correct grammatical and spelling errors. This is crucial to ensure clear communication of the research findings.

Answer 1: Thanks for your valuable suggestion. Our manuscript has been edited by AJE service with the verification code 8EDE-76A1-63BD-948C-319A.

Question 2. Clarifications regarding the inclusion and exclusion criteria of studies, particularly the rationale behind excluding studies that provide Kaplan-Meier survival curves instead of HR with 95%, are necessary.

Answer 2: Thanks for your valuable comment. We have added more description about the inclusion and exclusion criteria, especially the rationale behind excluding studies that provide Kaplan-Meier survival curves instead of HR with 95% CI as follows: “studies did not report the HRs with 95% CIs for survival, including studies providing Kaplan-Meier survival curves because of the significant bias during the calculation of HRs with 95% CIs from the Kaplan-Meier survival curves.” (Page 4, line 22-25)

Question 3. The association between sarcopenia and postoperative short-term outcomes should be explored or discussed as a limitation.

Answer 3: We have added this issue in the manuscript as one of the limitations: “Fourth, due to the lack of relevant data, we failed to explore the association between sarcopenia and postoperative short-term outcomes.” (Page 10, line 2-3)

Question 4. Detailed methodology and clarity in reporting, such as literature search strategy, study selection process, and handling of hazard ratios, are necessary for reproducibility and transparency.

Answer 4: Thanks for your valuable comment. The method part has been modified (Page 4, line 4 to Page 4, line 21)

Question 5. Include citations for statements concerning survival analysis and the types of included studies (prospective, retrospective, or RCT), as well as follow-up time specificity in table 1.

Answer 5: Thanks for your valuable comments. All these information have been updated and added in the manuscript. (Page 6, line 4; Table 1 and 2)

Question 6. Subgroup analyses based on predefined criteria and a discussion on publication bias are recommended to strengthen the paper's validity.

Answer 6: Thanks for your question. A discussion on publication bias has been added in the manuscript. “Fifth, significant publication bias was detected in our meta-analysis. Although the results of the fill-and-trim method indicated that “unpublished studies” did not affect the overall findings, more high-quality studies are still needed to verify our conclusions.” (Page 10, line 3-6) However, after careful team discussion, we deem that it is hard to conduct subgroup analyses based on predefined criteria. Therefore, we discussed this in the limitations part. “Secondly, some confounding factors such as the definition of sarcopenia including the cutoff values of SMI, PMV and PMI and adjuvant therapy exists in this meta-analysis and we are unable to conduct more detailed analysis due to the lack of original data” (Page 9, line 26-29) Meanwhile, subgroup analysis based on the sample size and ethnicity was further performed.

Question 7. Discrepancies in the numbers presented in tables regarding OS and DFS need to be addressed and corrected for consistency and clarity.

Answer 7: Dear editor, thanks for your question. Actually, as mentioned in the Table 1, some studies did not provide specific information about the neoadjuvant therapy, which were noted as “mixed”. Therefore, these studies were excluded during the subgroup analysis.

Question 8. All abbreviations, such as SMI, must be fully described upon their initial mention in the text, ensuring accessibility for all readers.

Answer 8: We have carefully checked and modified all abbreviations in this manuscript.

Response to Reviewer #1: 

Thanks for giving me the opportunity to review this work. In overall, this manuscript was well designed and written. It is a update meta-analysis investigating the prognostic value of sarcopenia in rectal cancer. I have few concerns about this work which would be addressed.

Question 1. English editing is recommended and some grammar errors exist in the manuscript.

Answer 1: Thanks for your valuable suggestion. Our manuscript has been edited by AJE service with the verification code 8EDE-76A1-63BD-948C-319A.

Question 2. Studies providing the Kaplan-Meier survival curves instend of HR with 95% were excluded? Why?

Answer 2: Thanks for your valuable question. In this meta-analysis, we excluded studies providing the Kaplan-Meier survival curves instead of HR with 95% CI because there was significant bias during the calculation of HRs with 95% CIs from Kaplan-Meier survival curves. In other words, the HRs with 95% CIs calculated based on Kaplan-Meier survival curves differ greatly from actual HRs with 95% CIs. Therefore, for the accuracy of results, we decided to exclude these studies.

Question 3. The authors did not identify the association between sarcopenia and postoperative short-term outcomes in rectal cancer patients. It's also a limitation.

Answer 3: We have added this issue in the manuscript as one of the limitations: “Fourth, due to the lack of relevant data, we failed to explore the association between sarcopenia and postoperative short-term outcomes. ” (Page 10, line 2-3)

Response to Reviewer #2: 

The authors included 15 studies and indicated the prognostic value of sarcopenia in rectal cancer. Congratulations on this successful manuscript. I have some minor suggestions for authors to further improve the quality of their work.

Question 1. I would recommend suggest the author carefully check and modify the manuscript and some spelling mistakes and grammatical mistakes could be noticed in this paper.

Answer 1: We have carefully checked and modified this manuscript. Meanwhile, our manuscript has been edited by AJE service with the verification code 8EDE-76A1-63BD-948C-319A.

Question 2. In the "literature search" part, detailed process should be described.

Answer 2: Detailed process has been described in the results part as follows “As shown in Figure 1, 764 records were found in these three databases and 101 duplicated records were removed. After reviewing the titles, 615 records were excluded. Then 25 publications were further excluded after reviewing the abstracts. After carefully reviewing full texts, 15 studies were eventually included in this meta-analysis” (Page 5, Line 25-29).

Question 3. "Thirteen studies explored the relationship between sarcopenia and OS among surgical rectal cancer patients." The references should be cited. The same suggestion for "Twelve studies manifested the association of sarcopenia with DFS among surgical rectal cancer patients." and "Only four studies explored the association of sarcopenia with postoperative CSS in rectal cancer patients."

Answer 3: This issue has been addressed and relevant references have been cited in the results part. (Page 6, line 11 to Page 7, line 9)

Question 4. What are the types of included studies? Retrospective or prospective? Or RCT?

Answer 4: All included studies were retrospective. (Page 6, line 4)

Question 5. In table 1, "mean" or "median" follow-up time should be specificly described for each studies.

Answer 5: The information about the follow-up time has been updated in the Table 1.

Response to Reviewer #3: 

In this study, the authors conducted a meta-analysis to evaluate the prognostic significance of preoperative sarcopenia in rectal cancer patients. Interestingly, they found that preoperative sarcopenia is associated with poor clinical outcomes in this disease. The manuscript's field is interesting, the introduction and discussion are well-structured; the result section is appropriate; however, there are some concerns, and major revision is expected.

Question 1. To enhance the transparency of the methodology, the authors should consider presenting the literature search strategy employed in various databases in a table. Furthermore, it is important for them to explicitly state the number of studies obtained from each database.

Answer 1: Thanks for your valuable question. We have added the Supplementary file 1 describing the literature search strategy employed in various databases. The number of studies obtained from each database was presented in Figure 1. Meanwhile, we have added this information in the results part (Page 5, line 26).

Question 2. For a more comprehensive description of the inclusion and exclusion criteria, it is advisable to provide a thorough explanation that encompasses all relevant aspects. This should include any limitations on language or excluding review articles.

Answer 2：Thanks for your valuable suggestion. The inclusion and exclusion criteria have been modified as request. (Page 4, line 11-25) 

Question 3. In the methods section, the authors should mention whether all hazard ratios (HRs) were provided in the included papers based on univariate or multivariate analysis (in table 1, add which one was for each study) or if they calculated the HRs based on the Kaplan-Meier curves.

Answer 3: We have added the description about the sources of HRs and 95% CIs in the statistical analysis part (Page 5, line 12-13) and also table 1. Studies providing Kaplan-Meier survival curves instead of HRs and 95% CIs were excluded in this meta-analysis.

Question 4. In the method section, on page 5, lines 1-2, please replace the word 'or' with 'otherwise' (or any suitable synonym) in the following sentence: 'The random effects model was applied; or the fixed effects model was applied.' The use of 'or' is incorrect.

Answer 4: The “or” has been changed to “otherwise”. (Page 5, line 11)

Question 5. I recommend doing subgroup analysis based on different criteria such as sample size (based on mean or median), Newcastle–Ottawa Scale score (<7 and >7), ethnicity (Asian and non-Asian), and so on.

Answer 5: Thanks for your valuable question. After careful team discussion, we have added subgroup analysis based on the sample size and ethnicity. (Table 2; Page 5, line 18-20; Page 7, line 4-6)

Question 6. In the discussion part, specially limitation section, the authors should mention and interpret the results regarding publication bias.

Answer 6: Thanks for your question. A discussion on publication bias has been added in the manuscript. “Fifth, significant publication bias was detected in our meta-analysis. Although the results of the fill-and-trim method indicated that “unpublished studies” did not affect the overall findings, more high-quality studies are still needed to verify our conclusions.” (Page 9, line 3-6)

Question 7. In the table 2, the overall number of studies for OS is 13. However, the number of studies for Chemoradiotherapy and NO subgroups are written 4 and 3, respectively. It is unclear. Because 4 plus 3 will be 7, not 13. So, please correct these numbers for OS and, also, DFS.

Answer 7: Dear reviewer, thanks for your question. Actually, as mentioned in the Table 1, some studies did not provide specific information about the neoadjuvant therapy, which were noted as “mixed”. Therefore, these studies were excluded during the subgroup analysis.

Question 8. Abbreviations such as SMI (Page 5) should be fully described when initially mentioned.

Answer 8: We have carefully checked and modified all abbreviations in this manuscript.

---

## [Decision Letter · Decision Letter 1]

6 Mar 2024

PONE-D-23-38922R1Predictive role of preoperative sarcopenia for long-term survival in rectal cancer patients: a meta-analysisPLOS ONE

Dear Dr. Su,

Thank you for submitting your manuscript to PLOS ONE. After careful consideration, we feel that it has merit but does not fully meet PLOS ONE’s publication criteria as it currently stands. Therefore, we invite you to submit a revised version of the manuscript that addresses the points raised during the review process.

We look forward to receiving your revised manuscript.

Kind regards,

Zubing Mei, MD,PH.D

Academic Editor

PLOS ONE

Journal Requirements:

**Additional Editor Comments:**

Most of the concerns have been adequately addressed. However, the reviewer still raised small issues that should be revised.

Reviewers' comments:

Reviewer's Responses to Questions

**Comments to the Author**

1. If the authors have adequately addressed your comments raised in a previous round of review and you feel that this manuscript is now acceptable for publication, you may indicate that here to bypass the “Comments to the Author” section, enter your conflict of interest statement in the “Confidential to Editor” section, and submit your "Accept" recommendation.

Reviewer #1: All comments have been addressed

Reviewer #3: All comments have been addressed

2. Is the manuscript technically sound, and do the data support the conclusions?

Reviewer #1: Yes

Reviewer #3: (No Response)

3. Has the statistical analysis been performed appropriately and rigorously? 

Reviewer #1: Yes

Reviewer #3: (No Response)

4. Have the authors made all data underlying the findings in their manuscript fully available?

Reviewer #1: Yes

Reviewer #3: (No Response)

5. Is the manuscript presented in an intelligible fashion and written in standard English?

Reviewer #1: (No Response)

Reviewer #3: (No Response)

6. Review Comments to the Author

Reviewer #1: The authors have addressed all comments. I do not have additional comments. I would suggest publication for this paper.

Reviewer #3: Thank you for replying appropriately to the comments.

I have noticed a minor issue in the Prisma flow diagram (figure 1). The total number of studies included in the meta-analysis has been written as n=25, but I guess it should be n=15 (the authors have written n=15 in the result part, page5/19, line 22). Please take this into consideration.

7. PLOS authors have the option to publish the peer review history of their article (what does this mean?). If published, this will include your full peer review and any attached files.

Reviewer #1: No

Reviewer #3: No

---

## [Author Response · Author response to Decision Letter 1]

6 Mar 2024

Response to Reviewer #3: 

I have noticed a minor issue in the Prisma flow diagram (figure 1). The total number of studies included in the meta-analysis has been written as n=25, but I guess it should be n=15 (the authors have written n=15 in the result part, page5/19, line 22). Please take this into consideration.

Answer: We are sorry for this mistake. The Figure 1 has been modified and reuploaded.

---

## [Editor Report · Decision Letter 2]

26 Apr 2024

Predictive role of preoperative sarcopenia for long-term survival in rectal cancer patients: a meta-analysis

PONE-D-23-38922R2

Dear Dr. Su,

We’re pleased to inform you that your manuscript has been judged scientifically suitable for publication and will be formally accepted for publication once it meets all outstanding technical requirements.

Kind regards,

Zubing Mei, MD,PH.D

Academic Editor

PLOS ONE
---

## [Editor Report · Acceptance letter]

1 May 2024

PONE-D-23-38922R2 

PLOS ONE

Dear Dr. Su, 

I'm pleased to inform you that your manuscript has been deemed suitable for publication in PLOS ONE. Congratulations! Your manuscript is now being handed over to our production team.

Kind regards, 

on behalf of

Dr. Zubing Mei 

Academic Editor

PLOS ONE